# Histone and DNA Methylation as Epigenetic Regulators of DNA Damage Repair in Gastric Cancer and Emerging Therapeutic Opportunities

**DOI:** 10.3390/cancers15204976

**Published:** 2023-10-13

**Authors:** Katia De Marco, Paola Sanese, Cristiano Simone, Valentina Grossi

**Affiliations:** 1Medical Genetics, National Institute of Gastroenterology—IRCCS “Saverio de Bellis” Research Hospital, Castellana Grotte, 70013 Bari, Italy; katia.demarco@irccsdebellis.it (K.D.M.); paola.sanese@irccsdebellis.it (P.S.); 2Medical Genetics, Department of Precision and Regenerative Medicine and Jonic Area (DiMePRe-J), University of Bari Aldo Moro, 70124 Bari, Italy

**Keywords:** gastric cancer, epigenetic alterations, methylation, histone methyltransferases, DNA damage repair, homologous recombination deficiency, PARP inhibitors

## Abstract

**Simple Summary:**

Histone and DNA methylations are widely studied epigenetic marks in gastric cancer. Methylation is involved in DNA damage repair (DDR) and can also regulate non-histone DDR proteins, thereby orchestrating DNA repair processes. Homologous recombination is the main pathway involved in the repair of double-strand breaks. Defects in the homologous recombination DNA repair system can cause homologous recombination deficiency (HRD). In this context, targeting DDR pathways has emerged as a promising cancer treatment strategy, with DDR inhibitors (especially PARP inhibitors) showing clinical success. In this review, we will discuss the alterations detected in histone and DNA methylation in gastric cancer and describe how these epigenetic processes affect DDR in gastric carcinogenesis. Moreover, we will dissect the applications of DDR inhibitors in the context of HRD and their promising role in gastric cancer treatment.

**Abstract:**

Gastric cancer (GC), one of the most common malignancies worldwide, is a heterogeneous disease developing from the accumulation of genetic and epigenetic changes. One of the most critical epigenetic alterations in GC is DNA and histone methylation, which affects multiple processes in the cell nucleus, including gene expression and DNA damage repair (DDR). Indeed, the aberrant expression of histone methyltransferases and demethylases influences chromatin accessibility to the DNA repair machinery; moreover, overexpression of DNA methyltransferases results in promoter hypermethylation, which can suppress the transcription of genes involved in DNA repair. Several DDR mechanisms have been recognized so far, with homologous recombination (HR) being the main pathway involved in the repair of double-strand breaks. An increasing number of defective HR genes are emerging in GC, resulting in the identification of important determinants of therapeutic response to DDR inhibitors. This review describes how both histone and DNA methylation affect DDR in the context of GC and discusses how alterations in DDR can help identify new molecular targets to devise more effective therapeutic strategies for GC, with a particular focus on HR-deficient tumors.

## 1. Introduction

Gastric cancer (GC) is the second highest cause of cancer mortality worldwide [1]. GCs can be categorized histologically using different classification systems; among these, Lauren’s classification is commonly used as a prognostic and pathogenic indicator for surgical strategy [2,3,4]. 

GC exhibits wide heterogeneity at the histopathological and molecular levels, resulting in a complex scenario for patients’ clinical management and prognosis [5]. The pathogenesis of GC is a classic example of gene–environment interaction, with diet, active tobacco smoking, and infection by Helicobacter pylori being the most common environmental factors related to gastric carcinogenesis [6]. While most GCs are considered sporadic, it is estimated that 1% to 3% are associated with inherited cancer predisposition syndromes [7]. GC can be linked to mutations in high-penetrance genes, such as *CDH1* [8], *SMAD4* [9], *STK11* [10], *APC* [11], *MLH1*, *MSH2*, *MSH6*, and *PMS2* [12], or in low-penetrance genes, such as *ATM* [13], *BLM* [14], *BRCA1/2* [15], and *TP53* [16]. In addition to genetic and environmental factors, epigenetic alterations also play a critical role in GC pathogenesis, as epigenetic regulation is associated with chromatin modifications such as DNA methylation and histone modifications, which affect multiple cellular processes, including gene expression and DNA damage repair (DDR) [17].

DDR is important for protecting against tumorigenesis [18]. During this process, histones undergo post-translational modifications (PTMs), leading to a more accessible chromatin configuration and facilitating the recruitment of DDR proteins to damaged DNA sites [18]. Methylation is a crucial histone PTM involved in DDR and can also regulate non-histone DDR proteins, thereby orchestrating DNA repair [18]. Several signaling cascades and over 450 related proteins are involved in DDR processes [19], with the repair pathway depending on the nature of the damage [20]. Single-strand breaks (SSBs) can be repaired by various mechanisms, including mismatch repair (MMR), base excision repair (BER), nucleotide excision repair (NER), and direct repair [20]. On the other hand, homologous recombination (HR) and non-homologous end joining (NHEJ) are the two main pathways involved in the repair of double-strand breaks (DSBs) [21].

Defects in the HR DNA repair system can cause homologous recombination deficiency (HRD), which results in impaired DSB repair; thus, malfunctions in this pathway are frequently associated with cancer [22]. An increasing number of defective genes playing a major role in HR are being identified in GC [23]. In this context, targeting DDR pathways has emerged as a promising cancer treatment strategy, with DDR inhibitors (especially PARP inhibitors (PARPis)) showing clinical success [23].

In this review, we will provide an overview of the alterations detected in histone and DNA methylation in GC and describe how these epigenetic processes affect DDR in gastric carcinogenesis. Then, we will summarize the main DNA repair pathways, with a particular focus on HR processes. Finally, we will outline the applications of DDR inhibitors in the context of HRD and their promising role in GC treatment.

## 2. Methylation: A Key Epigenetic Alteration in GC

GC is a complex and heterogeneous disease that develops from the accumulation of genetic and epigenetic alterations during patients’ lifetimes [24]. Epigenetic alterations are defined as heritable modifications in gene expression that occur without changes in the DNA sequence [25]. Numerous studies have revealed the importance of epigenetic alterations as driver events of tumorigenesis in both early and advanced stages of GC [5,26]. Indeed, these mechanisms can lead to altered gene expression patterns resulting in the activation of oncogenes and/or the inactivation of tumor suppressor genes [27].

Epigenetic alterations involve several processes that include histone, DNA, and RNA modifications [28]. They are commonly detected in human cancers, including GC, and can inactivate genes implicated in important cellular pathways such as DNA repair [29]. Research on epigenetics helped to elucidate the underlying mechanisms of GC initiation and progression [30]. The most extensively studied epigenetic alterations in GC are DNA and histone methylation events, which affect chromatin structure [28]. The methylation of chromatin components is a dynamic process that is essential for cell fate determination and genomic stability and is involved in gastric carcinogenesis due to its key role in the regulation of gene expression and DDR [31,32].

### 2.1. Histone Methylation and Its Role in DDR in GC

Histone proteins influence the higher-order structure of chromatin and play a major function in several processes in the cell nucleus, such as DNA replication, transcriptional activation or inhibition, and DNA accessibility to the repair machinery [33]. Histones are organized in octamers to wrap DNA into nucleosomes, which comprise two copies of each of the four canonical histone isotypes (H2A, H2B, H3, and H4) and histone H1 [33]. Methylation of lysine and arginine residues on histone tails regulates chromatin structure and is frequently dysregulated in cancer, including GC [34,35]. This mechanism affects not only gene expression but also histone–histone and histone–DNA interactions, thereby regulating the accessibility of DNA damage sites to repair complexes and the choice of the DNA repair pathway to be activated [5,25,28]. The chromatin landscape, with its dynamic structure and histone marks, influences the competition of repair factors on damaged DNA sites [25]. For example, upon a DSB, histone marks associated with open chromatin, such as H3K36 trimethylation, enable HR protein recruitment, since chromatin relaxation favors end resection; conversely, the NHEJ pathway is preferentially induced in regions of heterochromatin, because a compact structure and the presence of repetitive elements hinder the HR pathway [25].

The choice of the DNA repair pathway is also affected by other regulatory factors, such as R-loops. R-loops are three-stranded nucleic acid structures consisting of an RNA–DNA hybrid and a displaced non-template single-stranded DNA [36]. R-loops are important players in DNA replication and repair [37]. Altered R-loop levels have been detected in cancer cells, representing a source of DNA damage that triggers genome instability [38]. Indeed, while R-loops can be unwound by helicases or removed by RNase H, unresolved R-loops lead to DSBs through replication fork collapse or DNA cleavage by the XPG and XPF endonucleases [39]. Of note, R-loops can also be induced by DNA damage and act as major intermediates in DNA repair [36]. Several types of RNA–DNA hybrids have been shown to recruit DNA repair proteins to DSBs. In human cells, R-loops at DSBs can facilitate both HR and NHEJ repair processes; however, it seems that they predominantly promote the HR pathway because they enhance resection, an essential step of HR repair [36,40,41].

Moreover, R-loops play a role in the regulation of gene expression and chromatin structure and are involved in histone PTMs; indeed, they have been associated with H3K4 and H3K36 methylation, two marks of active transcription and chromatin decondensation [36,42]. On the other hand, R-loops can also promote heterochromatin assembly and chromatin compaction by mediating gene-repressive histone modifications, such as histone H3K9 dimethylation [36]. As such, R-loops can act as epigenetic marks that are read by chromatin remodelers and other proteins to induce changes in the chromatin state, therefore affecting gene transcription and DNA repair [40,43].

While histone methylation has been primarily characterized for its function in transcriptional regulation, it is also known to play a key role in DNA repair [18]. DNA damage-responsive histone methylation has been identified at several lysine residues of histone H3 and H4, including H3K4, H3K9, H3K27, H3K36, H3K79, and H4K20 [44]. These methylation marks can alter chromatin structure, regulating its accessibility to transcriptional and DNA repair proteins, depending on which histone residue is methylated [44]. Indeed, it has been shown that methylation of H4K5, H4K20, H3K9, di- and trimethylation of H3K27, and trimethylation of H3K79 are associated with a closed chromatin structure, whereas methylation of H3K4, H3K36, monomethylation of H3K27, and mono- and dimethylation of H3K79 are related to an open chromatin structure [45,46] (Figure 1).

The methylation status of these histones is regulated by methyltransferases and demethylases, some of which have been reported to be overexpressed in GC, suggesting that they could be targeted for therapeutic intervention [5]. The histone methyltransferase (HMT) EZH2 has been extensively studied in gastric carcinogenesis. It catalyzes H3K27 trimethylation and has been found overexpressed in GC, being associated with progression, malignancy, and poor prognosis [47,48,49,50,51,52,53]. Other HMTs that are involved in GC include SUV39H2, which is responsible for methyl marks that are implicated in DDR mechanisms, such as di- and trimethylation of H3K9, promoting HR repair [54,55,56]; SUV39H1, which catalyzes H3K9 methylation and is involved in the initiation, development, and migration of gastric carcinoma [55,57]; and DOT1L, which mediates H3K79 methylation and influences DDR. Moreover, DOT1L has been shown to regulate phosphorylation of the H2A histone variant H2AX (γH2AX) and HR repair in CRC cell lines [46,58,59].

Various members of the human SMYD family of HMTs have also been shown to play a role in GC. SMYD2 regulates transcription, inhibits the tumor suppressor proteins p53 and retinoblastoma protein (RB1), and enhances the poly (ADP-ribose) activity of the oncogenic protein PARP1 in cancer cells [60]. This lysine methyltransferase is specific for histones H3K36 and H3K4 and has been found overexpressed in GC [60,61]. SMYD3 can specifically methylate histones at H3K4, H4K5, and H4K20 and is involved in signal transduction. Its overexpression in GC is associated with poor prognosis, being implicated in the proliferation, migration, and invasion of GC cells [62,63,64,65,66]. Moreover, Fasano et al. recently showed that SMYD3 interacts physically with various critical players involved in cancer pathways (i.e., RPB1, BLM, p130, and AMPK), and these interactions were validated in GC cells [67]. SMYD5 regulates transcription, genome stability, DNA replication, and DNA repair; it is involved in H4K20 trimethylation and has been found upregulated in stomach adenocarcinoma [55].

The list of HMTs that are involved in GC is continuously growing. Hu et al. reported high expression levels of the histone-lysine N-methyltransferase EHMT2 (G9a) in GC tissues. G9a catalyzes H3K9 and H3K27 methylation and is associated with GC metastasis [68,69,70,71,72]. Moreover, G9a regulates HR repair in cancer cells by promoting the recruitment of RPA and RAD51 proteins to DSB sites [73]. The H3K9 methyltransferases SETDB1 and SETDB2 and the H4K20 methyltransferase SETD8 have also been found overexpressed in GC and are linked to carcinogenesis [55,74,75,76,77,78,79,80]. In addition, SETDB1 and SETD8 are involved in DDR; in particular, SETDB1 is essential for the completion of HR repair, whereas SETD8 affects the choice of the DSB repair pathway [81,82]. Furthermore, upregulation of the HMTs NSD2, which mediates H3K36 dimethylation, and PRDM12, which is responsible for H3K9 trimethylation, has been observed in stomach adenocarcinoma [55,83].

The main HMTs that have been reported to be overexpressed in GC are summarized in Figure 1.

Histone demethylation also plays an important role in gastric carcinogenesis. Indeed, several histone lysine demethylases (KDMs) have been found overexpressed in GC tissues. These include the H3K4 demethylases KDM1A [35,84,85,86], KDM5A [87,88,89,90], KDM5B [91,92], and KDM5C [93], which are associated with proliferation and metastasis, the H3K36 demethylases KDM2A and KDM2B, which promote cell growth and migration [35,94], and the H3K9 demethylases KDM4B [95,96] and KDM4C [35], which are involved in tumor growth and invasion. Moreover, an association has been reported between overexpression of the H3K27 demethylase KDM6B and GC development [97,98].

Aberrant expression of enzymes regulating histone methylation has been widely reported in GC. In this scenario, further investigations are warranted to better define their role in DDR processes and their potential use as therapeutic targets.

### 2.2. Methylation of Non-Histone DDR Proteins by HMTs in GC

In addition to playing a crucial role in chromatin remodeling, HMTs regulate the methylation of non-histone substrates such as DDR proteins, leading to changes in their activity, interaction, or subcellular localization [99]. In particular, some HMTs that are overexpressed in GC are also involved in lysine methylation of non-histone DDR proteins, such as the tumor suppressor p53, which is regulated by several PTMs [100]. p53 is an important player in DDR, as it regulates checkpoint activation, induces cell cycle arrest and apoptosis, and blocks DNA repair [101]. p53 is methylated by the HMT SMYD2 at lysine K370. This methylation mark represses p53 activity and impairs its ability to bind to the promoters of target genes [102]. Moreover, p53 is methylated at other lysine residues by two further HMTs that have been found upregulated in GC; specifically, it is dimethylated at K373 by G9a and monomethylated at K382 by SET8, both of which have a repressive effect [103,104]. SMYD2 is also responsible for the methylation of the tumor suppressor RB1 at lysines K860 and K810 and for the methylation of PARP1 at lysine K528 [105,106]. The RB1 protein is involved in cell cycle progression and DDR, where it binds to components of the NHEJ pathway [107]. It has been shown that SMYD2-dependent methylation of RB1 at K810 promotes cell cycle progression in cancer cells [105,107]. Recruitment of the PARP1 enzyme to DNA damage sites is regulated by SMYD2 methylation at K528, a methyl mark that enhances PARP1 activity in cancer cells [105]. Proliferating cell nuclear antigen (PCNA) has been found methylated at K248 by the HMT SETD8, which leads to the stabilization of PCNA expression; this is an important event in DNA replication and repair because PCNA acts as a binding platform for several enzymes involved in these processes [108,109]. Wang et al. showed that SMYD3 directly methylates EZH2 at K421, which promotes GC progression [110], and EZH2 seems to play a role in DDR [111]. Furthermore, since SMYD3 has been reported to modulate an ATM-related pathway in GC cells, it may also be involved in DDR processes in this cancer type, as has already been observed in other tumors, thus gaining ground as an interesting therapeutic target [112,113,114].

Lysine methylation of non-histone proteins recently emerged as an important regulator of DNA damage-related processes. As such, in-depth characterization of the complex interplay between epigenetic factors and non-histone DDR proteins may be useful to devise innovative anticancer strategies.

### 2.3. Effects of Promoter Methylation on the Expression of Major DNA Repair Genes in GC

Epigenetic changes modify chromatin structure in various ways, including at the DNA level [115]. DNA methylation is catalyzed by DNA methyltransferases (DNMTs) [116]. The main members of this enzyme family are DNMT1, DNMT3A, and DNMT3B [116]. DNMT1 is responsible for the maintenance of the DNA methylation status during DNA replication, while DNMT3A and DNMT3B are de novo methyltransferases contributing to the regulation of the DNA methylation pattern [116]. Overexpression of DNMTs has been shown to play a role in GC and various other tumor types by turning off the expression of several genes, including tumor suppressors [117].

In GC, increased expression of DNMT1 compared to para-cancerous and normal tissues has been reported at both the mRNA and protein levels, with DNMT1 upregulation being associated with GC risk and worse prognosis [5,118,119,120,121,122]. Moreover, it has been found that DNMT1 overexpression is correlated with stomach tumor localization, DNMT3 overexpression is correlated with the tumor-node-metastasis (TNM) score, and the concomitant overexpression of DNMT1 and DNMT3A is significantly associated with lymph node metastasis [5,118,119,120,121,122,123].

Methylation differences have been observed in GC in a variety of gene classes involved in important cellular pathways, including DNA repair [117]. Methylation of CpG islands often occurs at promoter regions and leads to alterations in chromatin structure with a decrease or an increase in the rate of transcription [117]. Promoter hypermethylation occludes the binding sites of transcription factors, leading to the inhibition of DNA repair gene expression [124]. This mechanism is schematically represented in Figure 2.

Bernal et al. detected *BRCA1* promoter hypermethylation in GC, resulting in decreased BRCA1 mRNA and protein levels [125]. In normal tissue, BRCA1 forms complexes with BRCA2 and other proteins to activate DSB repair and initiate HR, contributing to the maintenance of genomic integrity [126]. This suggests that hypermethylation of the *BRCA1* promoter is involved in gastric carcinogenesis.

Aberrant methylation of genes involved in the DNA MMR pathway has also been observed in GC [5]. Importantly, MMR promotes genome stability in all organisms by correcting DNA base mismatches and insertion or deletion loops that occur during DNA replication, repair, and recombination [127]. In GC, DNA MMR genes such as *MLH1*, *MSH2*, and *PMS2* have been found hypermethylated in their promoter region, resulting in the loss of protein expression [128,129]. The MMR protein MLH1 regulates EXO1 nuclease activity during DNA repair, and its loss causes DNA hyper-excision, resulting in chromosomal instability and cytosolic DNA accumulation; moreover, MLH1 has been shown to play a pivotal role in DNA strand break repair because it is recruited to DNA sites damaged by various genotoxic agents, thus acting as a damage sensor [130,131]. Repair is accomplished by MSH2, which binds MSH3 and MSH6 to form the MutS complex, and by PMS2, which forms the MutL complex with MLH1, PMS1, and MLH3 [132]. These complexes recognize errors in the genome sequence and start the signaling process to replace damaged DNA through the action of DNA polymerase δ and DNA ligase I [132]. Alterations in the expression of these genes can thus result in the accumulation of mutations [132]. Promoter methylation of other genes involved in DDR has also been shown to play a role in GC. Methylation of the promoter of the DNA repair enzyme O6-alkylguanine DNA methyltransferase (*MGMT*), which protects the DNA from mutations caused by alkylating agents, results in gene silencing and loss of function, contributing to cancer development [133]. Likewise, hypermethylation of the *XRCC1* gene promoter was found to be linked to loss of protein expression [134]. The XRCC1 protein is the key enzyme of the BER pathway, which recognizes DNA SSBs, binds to the DNA, and recruits other components of the repair pathway, such as PARP1 [19,134]. Thus, *XRCC1* promoter hypermethylation can cause an increase in DNA damage and the accumulation of genetic abnormalities [134].

Importantly, recent observations suggest that specific DNMTs are directly involved in DDR [135]. In particular, DNMT1 is rapidly and transiently recruited to regions of DNA DSBs via its interaction with PCNA, where it colocalizes with γH2AX, a DSB marker [136]. In addition to PCNA, DNMT1 also interacts with other components of the DDR machinery, including CHK1 and the 9-1-1 complex, which are essential for DNMT1 recruitment to DNA damage sites [137,138]. However, since DNMT1 is recruited only transiently to these sites, presumably before DNA re-synthesis completion, it seems that its main function is not the restoration of DNA methylation patterns [135]. It has thus been proposed that DNMT1 may mediate chromatin relaxation to enable access of DNA repair factors and/or contribute to the response to specific DNA damage events [135,138].

The involvement of DNMT1 in DDR has important implications for therapeutic strategies aimed at targeting DNA methylation in tumor cells.

There is strong evidence that DNA methylation affects DDR processes, with aberrant methylation patterns at the promoter regions of DDR genes leading to alterations in gene expression and defects in cellular repair pathways. DNMTs thus represent an interesting area of investigation for targeted therapies.

## 3. HR: The Predominant DSB Repair Pathway

As indicated above, the two major repair pathways involved in DSB repair are HR and NHEJ. HR is a high-fidelity pathway that requires a DNA template for the repair process to restore the original sequence, while NHEJ is intrinsically error prone because it is template independent and results in the direct ligation of DNA ends [139].

In normal cells, DSBs are preferentially repaired by HR rather than by NHEJ [140]. In the HR pathway, which occurs during the late S to G2 phase of the cell cycle, DSBs are first sensed by the MRE11-RAD50-NBS1 (MRN) complex, which binds to the broken DNA ends and loads the BLM helicase and the EXO1 nuclease to perform bulk 5′-3′ DNA resection [141]. The MRN complex also activates the key signaling kinase ATM at DSB sites; once activated, ATM induces cell cycle arrest via CHK2 activation and facilitates repair through the phosphorylation of various downstream proteins taking part in DDR [13]. In addition, the MRN complex allows single-stranded DNA (ssDNA) overhangs to be covered by the RPA protein, which suppresses further resections and facilitates the assembly of factors involved in the HR pathway, such as the RAD51 recombinase [142]. The ATR kinase is then recruited to RPA-coated ssDNA as a result of the interaction between RPA and ATRIP, and ATR-ATRIP complexes phosphorylate the CHK1 protein to arrest the cell cycle and protect stalled replication forks [141]. Following cell cycle arrest, RAD51, which is activated by BRCA2–PALB2 interaction, localizes into the nucleus and displaces RPA from the 3′ overhangs to form presynaptic filaments; subsequently, the resected ends undergo DNA synthesis and ligation using the intact chromatid as a template [143].

### HRD as a Common Feature in GC

Accumulated evidence indicates that mutations in genes involved in HR DNA repair are common occurrences in GC [144]. Defects in the HR pathway cause cells to rely on more error-prone DNA repair pathways such as NHEJ, leading to genomic instability and cancer development [145]. BRCA1 and BRCA2 are crucial components of the HR pathway and have been identified as major tumor susceptibility genes since pathogenic mutations in these genes result in defective HR repair [146]. In GC, BRCA1 deficiency is significantly associated with patients with diffused Lauren type, higher tumor grades, and advanced clinical stage; these patients live significantly shorter than those with positive expression of BRCA1, indicating that loss of *BRCA1* can serve as a prognostic marker [145]. Impairment of *BRCA1* gene function in GC has also been associated with epigenetic modifications, namely promoter methylation, causing loss of protein expression and thus affecting DNA repair capacity [125].

Alterations in other proteins involved in the HR pathway can have the same effect as *BRCA1* and *BRCA2* pathogenic mutations, leading to HRD [146]. ATM is essential for HR repair in response to DSB damage. *ATM* gene mutations have been detected in GC cell lines, and decreased ATM mRNA and protein levels have been found in tumoral tissue compared to normal samples; moreover, low levels of phosphorylated ATM were found to be significantly correlated with poor differentiation, lymph node metastasis, and decreased 5-year survival [147]. Zhang et al. reported that a common feature in GC is the loss of the ATR protein, another major sensor of DNA damage [145]. Mutations in HR genes, particularly *BRCA1*, *PALB2*, and *RAD51C*, which are important to maintain genome integrity in response to DNA damage, have been shown to increase GC risk [19,148]. CDK12 participates in DNA repair and contributes to cell division fidelity and genomic stability after DNA damage; its mutation has been reported in several malignancies, and decreased CDK12 expression is associated with a worse prognosis in GC [149]. Various other genes are recurrently altered in GC, albeit at a low frequency, including some coding for proteins involved in the regulation of the MRN complex, which recognizes DNA damage and triggers DDR [23,150].

Mutations in HR genes that result in impaired DNA DSB repair play an important role in GC progression. Indeed, HRD status has been associated with GC prognosis, enabling the prediction of clinical outcomes for specific patients [22]. The correlations identified between alterations in DDR genes or proteins and GC are summarized in Table 1.

## 4. Current Therapeutic Strategies Targeting DDR Proteins in GC

Cancer cells with a deficiency in the HR pathway will be more dependent on SSB repair mechanisms, which opens new possibilities for therapeutic interventions [140]. PARP is a nuclear enzyme acting as an SSB sensor protein and binds to SSBs to enable effective DNA repair [151]. In addition, PARP is involved in the activation of ATM, which is essential for HR, and in the inactivation of DNA-dependent protein kinases, which play an important role in the NHEJ pathway [152]. PARP inhibition leads to increased accumulation of SSBs that cannot be repaired and are converted into DSBs, which need to be repaired by HR; however, in HR-deficient cancer cells, DSBs cannot be repaired by this pathway, leading to a significant increase in genomic instability, with the newly formed DSBs becoming lethal for the cell [153]. This mechanism, in which the inactivation of one of two alternative cellular pathways is compatible with cell survival, but their concurrent inactivation leads to cell death, is termed “synthetic lethality” and accounts for the therapeutic success of PARPis [154]. PARPis inhibit PARP enzymatic activity by competing with NAD+ for binding to the PARP catalytic domain and producing conformational changes, thus preventing NAD+ utilization on PARP [152]. This prevents auto-poly-ADP-ribosylation and PARP release from the DNA lesion site, resulting in PARP being trapped on the DNA [140]. PARP–DNA complexes can lead to the accumulation of unrepaired SSBs and the stalling or collapsing of replication forks, causing the generation of DSBs, which represent a more deleterious damage; in this scenario, tumor cells having an underlying defect in HR will be unable to repair the DSBs, ultimately resulting in cell death [152]. Recent studies also reported other mechanisms of action for PARPis [140,155,156]. Since PARP can inhibit the NHEJ pathway, the use of PARPis promotes NHEJ, which is a less accurate repair mechanism, leading to unbearable levels of genomic instability that will elicit a tumoricidal effect [155]. Moreover, it has been shown that PARPis decrease the repair efficiency of the mutagenic microhomology-mediated end joining (MMEJ) or alternative end joining (Alt-EJ) pathway [140]. HR factors suppress MMEJ after DSB resection; however, in HR-deficient tumor cells, DSB repair can involve the MMEJ pathway, which requires PARP for efficient recruitment of the trans-disease polymerase (POLQ) to DSBs [140]. Therefore, PARPis will block the Alt-EJ pathway, promoting the death of HR-deficient tumor cells [155] (Figure 3).

Based on this evidence, PARP is considered an important therapeutic target for the development of cancer treatments, especially for *BRCA*-mutated tumors. Indeed, the clinical success of PARPis has brought new hope for synthetic lethal antitumor therapies, which represent a promising next-generation strategy targeting DDR [19]. Olaparib is the first PARPi that has been introduced in clinical practice. It has been tested as a monotherapy or in combination with other therapeutic agents in different solid tumor types, including GC, and in particular in a subset of patients with *BRCA* mutations [153,157,158]. Other PARPis, such as veliparib, talazoparib, niraparib, pamiparib, and rucaparib, are currently being investigated in clinical trials for GC treatment, alone or combined with other drugs [159,160,161,162,163].

HRD and the activity of HR genes are important determinants of therapeutic response to PARPis. Several proteins are involved in the HR repair pathway, and their alteration may render tumor cells more susceptible to PARPi monotherapy when HR repair is impaired [153]. In settings in which PARPi monotherapy has shown limited efficacy, combined strategies with PARPis can be devised to optimize efficacy and extend PARPi-based therapy to more tumor types with different molecular profiles [164].

In the wake of the clinical success of PARPis, other DDR inhibitors have been developed. AZD0156 and AZD6738 are potent and selective inhibitors of ATM and ATR, respectively [165]. They are both being tested in clinical trials as a monotherapy or in combination with the PARPi olaparib and other DNA-damaging agents [166,167]. Interestingly, recent studies have shown that the CHK1 inhibitor LY2606368 induces DNA damage and inhibits cancer cell proliferation [19]. Moreover, the WEE1 inhibitor AZD1775 has been shown to significantly inhibit proliferation and induce apoptosis and cell cycle arrest in GC cells, either alone or in combination with other DDR inhibitors [19,168,169].

Although further research is needed to validate the efficacy of DDR inhibitors in clinical settings, this approach may open new avenues for the development of targeted therapies for GC and other malignancies.

## 5. Conclusions and Future Directions

An increasing number of studies have shown that epigenetic alterations play a major role in DNA repair mechanisms. Histone methylation and DNA methylation are the two most widely studied epigenetic marks in GC, and multiple histone methyltransferases and demethylases have been found altered in GC, emerging as potential therapeutic targets. Indeed, these epigenetic modifications regulate the repair process of DNA damage, influencing the access of the repair machinery and the choice of the repair proteins on damaged sites. In the context of cancer, this function is particularly important as it affects how cells can respond to internally or externally induced DNA damage. Inhibitors of core proteins involved in DDR are promising for their potential to selectively target cancer cells; in this light, DDR inhibitors, especially PARPis, have shown to be effective when used in monotherapy as part of a synthetic lethality strategy in HR-deficient GC cells. In addition, PARPis may also prove successful in combination with other therapeutics. Therefore, identifying other biomarkers of HR deficiency to predict the response to DDR inhibitors in GC is highly desirable.

## Figures and Tables

**Figure 1 cancers-15-04976-f001:**
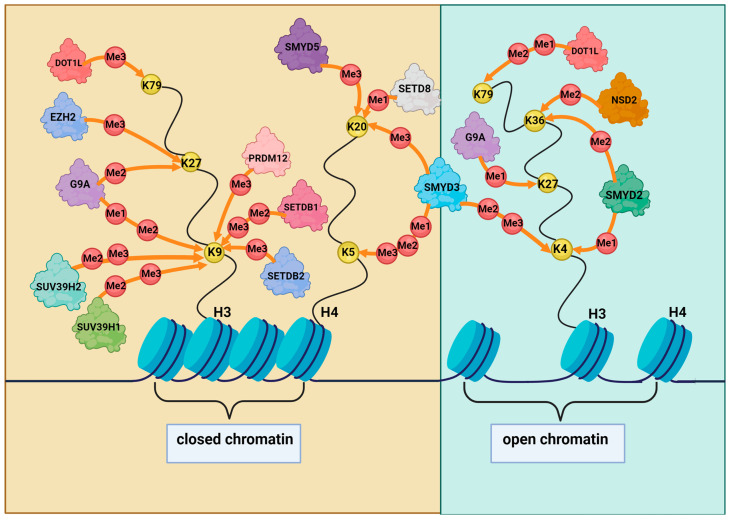
Schematic representation of the mechanism of action of the histone methyltransferases (HMTs) that have been found overexpressed in GC. HMTs orchestrate chromatin configuration and accessibility to the DNA repair machinery by adding methyl marks on histone lysine residues. Created with Biorender.com.

**Figure 2 cancers-15-04976-f002:**
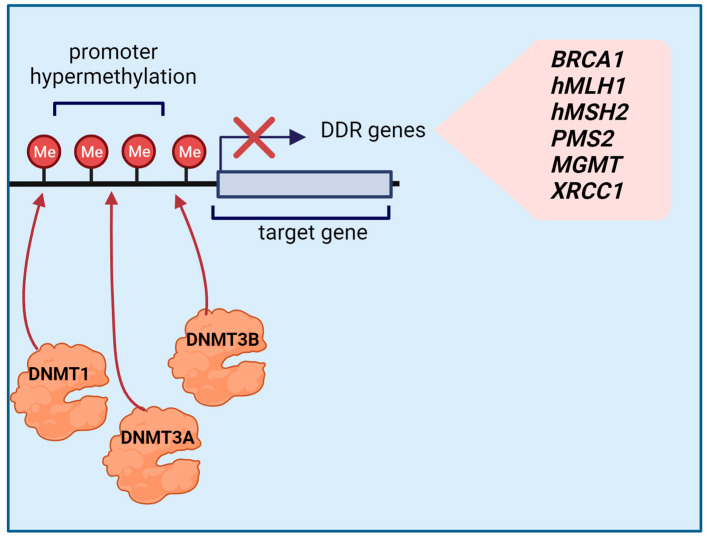
Schematic representation of the mechanism of action of the DNA methyltransferases (DNMTs) that have been found overexpressed in GC. DNMT-mediated methylation of CpG islands in the promoter regions of genes involved in DNA repair (pink box) affects the rate of transcription. Promoter hypermethylation occludes the binding sites of transcription factors, leading to the inhibition of gene expression. Created with Biorender.com.

**Figure 3 cancers-15-04976-f003:**
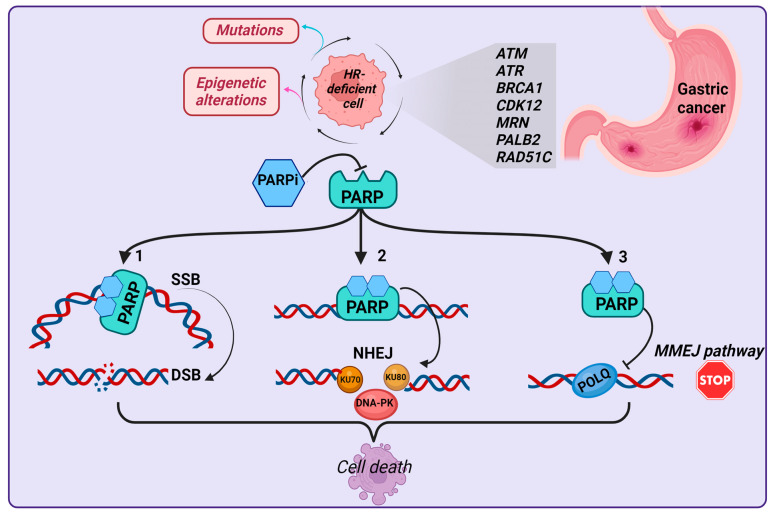
Schematic representation of the three mechanisms of action of PARPis in HR-deficient cells. In the first mechanism, PARP is trapped in a complex with the DNA, and SSBs are transformed into DBSs, ultimately causing cell death. In the second mechanism, PARPis enhance the NHEJ repair pathway, with the resulting genomic instability leading to cell death. In the third mechanism, PARPis inhibit the recruitment of POLQ, which is required for efficient DNA repair, and again the resulting genomic instability leads to cell death. Abbreviations: HR: homologous recombination; SSB: single-strand break; DSB: double-strand break; PARPi: PARP inhibitor; NHEJ: non-homologous end joining; MMEJ: microhomology-mediated end joining. Created with Biorender.com.

**Table 1 cancers-15-04976-t001:** Correlations between altered DDR genes/proteins and GC.

Altered DDR Gene or Protein	Function in HR	Study Design	Correlation	References
BRCA1	Crucial component of the HR pathway, it recruits BRCA2 complexes to DSB sites	In vivo	Has been associated with reduced survival time and patients with diffused Lauren type, higher tumor grades, and advanced clinical stage	[146,148]
ATM	Key player that starts the HR pathway by phosphorylating various DDR proteins	In vitro and in vivo	Has been linked to poor differentiation, lymph node metastasis, and decreased survival	[149,150]
ATR	Important kinase that orchestrates the HR repair pathway	In vivo	May promote cancer development	[146]
PALB2	BRCA2 complex interactor	In vivo	Potential valuable prognostic marker	[19,151]
RAD51C	BRCA2 interactor that plays a pivotal role in presynaptic filament assembly	In vivo	Potential valuable prognostic marker	[19,151]
CDK12	Affects DNA repair and contributes to the maintenance of genomic integrity	In vitro and in vivo	Has been associated with worse prognosis	[152]
MRN	One of the first sensors and responders to DSBs	In vivo	Has been associated with cancer development and worse overall patient survival	[146,153]

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
