# Peer review of "Histone and DNA Methylation as Epigenetic Regulators of DNA Damage Repair in Gastric Cancer and Emerging Therapeutic Opportunities"

_cancers, 2023, doi:10.3390/cancers15204976_

Round 1

Reviewer 1 Report

The review “Epigenetic alterations as regulators of DNA damage repair in gastric cancer and emerging therapeutic opportunities” is devoted to the interaction of DNA methylation and histone proteins with DNA damage repair in gastric cancer. The topic of DNA damage repair and application of PARP inhibitors has been well studied and actively discussed recently. The manuscript is properly written and organized, and contains nice visual illustrations. However, I have several remarks.

- The title of the review is very comprehensive, whereas the authors described only methylation out of all the epigenetic changes. It would be better to modify the title or expand the description of epigenetic changes, including histone acetylation and the effect of ncRNA.

- The review contains a lot of general, well-known information. The introduction part could be slightly reduced.

- Only the use of PARP inhibitors was described as a therapeutic approach. To increase the novelty of the review, the authors could add some additional approaches and drugs that affect the DNA damage repair.

Therefore, I recommend the review for publication in the Cancers journal after minor revision.

-

Author Response

Dear Editor,

we are pleased to submit the amended version of our work “Epigenetic alterations as regulators of DNA damage repair in gastric cancer and emerging therapeutic opportunities” (cancers-2486955), which we would like to have considered for publication in Cancers as part of the special issue “Drug Resistance in Gastrointestinal Cancer”. We addressed below all the reviewers’ comments by responding to their observations and by clarifying/adding sentences in the text in accordance with their suggestions.

Reviewer 1:

Comments and Suggestions for Authors

The review “Epigenetic alterations as regulators of DNA damage repair in gastric cancer and emerging therapeutic opportunities” is devoted to the interaction of DNA methylation and histone proteins with DNA damage repair in gastric cancer. The topic of DNA damage repair and application of PARP inhibitors has been well studied and actively discussed recently. The manuscript is properly written and organized, and contains nice visual illustrations. However, I have several remarks.

We thank the Reviewer for this comment. Below is a detailed point-by-point response to their remarks.

- The title of the review is very comprehensive, whereas the authors described only methylation out of all the epigenetic changes. It would be better to modify the title or expand the description of epigenetic changes, including histone acetylation and the effect of ncRNA.

We are grateful to the Reviewer for this observation. Since the purpose of this review was to provide an overview of the alterations detected in histone and DNA methylation in gastric cancer, we changed the title to “Histone and DNA methylation as epigenetic regulators of DNA damage repair in gastric cancer and emerging theraputic opportunities”.

- The review contains a lot of general, well-known information. The introduction part could be slightly reduced.

In accordance with the Reviewer’s suggestion, we slightly reduced the Introduction section.

- Only the use of PARP inhibitors was described as a therapeutic approach. To increase the novelty of the review, the authors could add some additional approaches and drugs that affect the DNA damage repair.

Therefore, I recommend the review for publication in the Cancers journal after minor revision.

We thank the reviewer for this comment as it prompted us to include an interesting additional topic in our review. In this amended version of the manuscript, we described recent progress in the field of DNA damage repair-related drugs in section 4, whose title we changed to “Current therapeutic strategies targeting DDR proteins in GC”.

Reviewer 2 Report

The manuscript "Epigenetic alterations as regulators of DNA damage repair in gastric cancer and emerging therapeutic oppurtunities" is a review that is focused on how methylation events regulate DNA repair and genome stability.

The manuscript is well written and easy to understand.

All figures presented are required.

The topic of the manuscript is potentially exciting.

Enthusiasm for this manuscript is brought down by the fact that as written the manuscript is not very novel or interesting. The authors begin the manuscript by introducing several factors that regulate cellular methylation and then simply fall back into how these regulators may influence known DNA repair proteins. 

It appears that the authors are missing the chance to discuss how these regulators play a role in DNA repair themselves. Factors such as Dotl1 and the DMNT's have been shown to play roles in the DNA repair process outside of the expression of known DNA repair proteins. DMNT's have been shown to be recruited to double strand break sites and change the DNA methylation status around double strand. It would be a much more interesting and novel review to discuss how methylation at DSB site influences DNA repair and genome stability.

In addition, the authors do not discuss the connection between R-loop formation, methylation patterns and DNA damage/repair. This is a timely topic, that would add additional novelty and interest to this review.

Author Response

Dear Editor,

we are pleased to submit the amended version of our work “Epigenetic alterations as regulators of DNA damage repair in gastric cancer and emerging therapeutic opportunities” (cancers-2486955), which we would like to have considered for publication in Cancers as part of the special issue “Drug Resistance in Gastrointestinal Cancer”. We addressed below all the reviewers’ comments by responding to their observations and by clarifying/adding sentences in the text in accordance with their suggestions.

Reviewer 2:

Comments and Suggestions for Authors

The manuscript "Epigenetic alterations as regulators of DNA damage repair in gastric cancer and emerging therapeutic oppurtunities" is a review that is focused on how methylation events regulate DNA repair and genome stability.

The manuscript is well written and easy to understand.

All figures presented are required.

The topic of the manuscript is potentially exciting.

Enthusiasm for this manuscript is brought down by the fact that as written the manuscript is not very novel or interesting. The authors begin the manuscript by introducing several factors that regulate cellular methylation and then simply fall back into how these regulators may influence known DNA repair proteins.

It appears that the authors are missing the chance to discuss how these regulators play a role in DNA repair themselves. Factors such as Dotl1 and the DMNT's have been shown to play roles in the DNA repair process outside of the expression of known DNA repair proteins. DMNT's have been shown to be recruited to double strand break sites and change the DNA methylation status around double strand. It would be a much more interesting and novel review to discuss how methylation at DSB site influences DNA repair and genome stability.

We are grateful to the Reviewer for raising this point. In accordance with their suggestion, in this amended version of the manuscript, we covered this important topic in section 2.3.

In addition, the authors do not discuss the connection between R-loop formation, methylation patterns and DNA damage/repair. This is a timely topic, that would add additional novelty and interest to this review.

We thank the Reviewer for this comment. In this amended version of the manuscript, we included an additional part in section 2.1 to discuss this topic as suggested.

Reviewer 3 Report

The title encompasses two concepts: that of epigenetic modifications in gastric cancers, and that of the actionability of these modifications. The term "gastric cancer" is imprecise, as it is evident that this article does not address gastric lymphomas or neuroendocrine tumors. Epigenetic modifications are frequent events in gastric carcinogenesis, and their roles are still unknown for some of them. These modifications pose multiple issues: their functions in the process of carcinogenesis, their place in recent clinico-pathological classifications combining genetic and (see TCGA molecular classification) and epigenetic data, their role in treatment resistance, and last but not least, their actionability in cancer treatment perspectives. This shows that the subject is rich.

Unfortunately, the chosen format is that of a catalog, which overlooks several important questions. Firstly, the focus is not specifically on gastric oncology or even on epigenetics. In fact, somatic mutations are given as much importance in the text as epigenetic events (see, for example, paragraph 3 on HR and 4 on PARPis). What about the frequency of BRCA mutations in gastric cancers? In these paragraphs, the focus should be on epigenetic events, e.g. the role of BRCA methylation in gastric cancers. Unfortunately, nothing is said about epigenetic modifications associated with PARP inhibitor resistance. The focus on "synthetic lethality" belongs to the realm of general oncology and does not merit extensive development.

Furthermore, some epigenetic alterations are not contextualized. It is important to discuss the data on CpG hypermethylation (including both promoter and non-promoter CpG islands) and universal CDKN2A promoter hypermethylation in the context of EBV-positive gastric cancers. Similarly, while MLH1 silencing is mentioned, nothing is said about its integration into the MSI phenotype and its clinico-pathological, prognostic, and therapeutic consequences (clinical trials).

No specific comment

Author Response

Dear Editor,

enclosed please find a revised version of manuscript entitled “Histone and DNA methylation as epigenetic regulators of DNA damage repair in gastric cancer and emerging therapeutic opportunities”, which we would like to have considered for publication in Cancers.

Please, note that the editorial office sent us a revision request for the manuscript on 7 August 2023 (major revision) therefore we downloaded the comments of 3 referees. Thus, we tried to upload the revised version together with our responses to the reviewers (3 referees) on 25th September (editorial deadline for resubmission) and we unexpectedly found a new referee’s  comments uploaded on 8 August 2023 2:55 pm (after editorial and official revision request).

We believe that the reviewers' suggestions (3 referees) have been very helpful in improving the manuscript and we hope that our amended manuscript can satisfy the overall reviewers request.

Gastric cancer (GC), one of the most common malignancies worldwide, is a heterogeneous disease developing from the accumulation of genetic and epigenetic changes. Epigenetic alterations play a critical role in GC pathogenesis, as epigenetic regulation is associated with chromatin modifications such as DNA methylation and histone modifications, which affect multiple cellular processes, including gene expression and DNA damage repair (DDR). During DDR, histones undergo post-translational modifications (PTMs), leading to a more accessible chromatin configuration and facilitating the recruitment of DDR proteins to DNA damaged sites. Methylation is a crucial histone PTM involved in DDR and can also regulate non-histone DDR proteins, thereby orchestrating DNA repair. Several DNA damage repair mechanisms have been recognized so far, with homologous recombination (HR) being the main pathway involved in the repair of the double-strand breaks (DSBs). Defects in the DNA repair system can cause homologous recombination deficiency (HRD), which results in impaired DSB repair; thus, malfunctions in these pathways lead to deleterious consequences and are frequently associated with cancer. An increasing number of defective HR genes are emerging in GC, resulting in the identification of important determinants of therapeutic response to PARP inhibitors (PARPis).

Considering all of the above, our work is an overview of the alterations detected in histone and DNA methylation in GC and describe how these epigenetic processes affect DDR in gastric carcinogenesis. Moreover, we summarize the main DNA repair pathways, with a particular focus on HR processes, and the applications of PARPis in the context of HRD and their promising role in GC treatment.

We believe that our work may attract the interest of multiple readers, from molecular and cellular biologists to clinicians, both in the academic community and in the pharmaceutical industry.

We look forward to your response.

Sincerely,

Cristiano Simone

Valentina Grossi

Reviewer 4 Report

De Marco et al. focused on the molecular and clinical importance of epigenetic alteration regarding DNA damage repair (DDR) in gastric cancer (GC). The text of this review article is of interest and is well-summarized about epigenetic alterations of DDR in GC, but their figures do not show the importance/specificity of epigenetics in GC.

Comments:

1)    I think that epigenetic alterations of DDR genes and clinicopathological significance of GC should be summarized in new Table.

2)    Figure 2 does not provide new information to readers. It is a general figure of DNA methylation in cancer. From this figure, what do the authors want to provide the information of epigenetics in GC? 

3)    Figure 3 show the general molecular mechanism of PARP in HR-deficient tumor cells and we can find it in many review article easily. The authors should modify the figure to new version including epigenetic alteration.

Author Response

Dear Editor,

we are pleased to submit the amended version of our work “Epigenetic alterations as regulators of DNA damage repair in gastric cancer and emerging therapeutic opportunities” (cancers-2486955), which we would like to have considered for publication in Cancers as part of the special issue “Drug Resistance in Gastrointestinal Cancer”. We addressed below all the reviewers’ comments by responding to their observations and by clarifying/adding sentences in the text in accordance with their suggestions.

Reviewer 4:

Comments and Suggestions for Authors

De Marco et al. focused on the molecular and clinical importance of epigenetic alteration regarding DNA damage repair (DDR) in gastric cancer (GC). The text of this review article is of interest and is well-summarized about epigenetic alterations of DDR in GC, but their figures do not show the importance/specificity of epigenetics in GC.

Comments:

1)    I think that epigenetic alterations of DDR genes and clinicopathological significance of GC should be summarized in new Table.

We are grateful to the Reviewer for this comment. In this amended version of the manuscript, we summarized this information in new Table 1 as suggested.

2)    Figure 2 does not provide new information to readers. It is a general figure of DNA methylation in cancer. From this figure, what do the authors want to provide the information of epigenetics in GC?  

We thank the Reviewer for this observation. In this amended version of the manuscript, we reorganized Figure 2 to focus on the overexpression of DNA methyltransferases resulting in promoter hypermethylation of genes involved in DNA repair in gastric cancer.

3)    Figure 3 show the general molecular mechanism of PARP in HR-deficient tumor cells and we can find it in many review article easily. The authors should modify the figure to new version including epigenetic alteration.

We thank the Reviewer for this useful comment. In this amended version of the manuscript, we included epigenetic alterations in Figure 3 as suggested. 

Round 2

Reviewer 2 Report

The authors appeared to have responded to the previous comments of this review. Thus, this is a much more interesting version of this manuscript.

Reviewer 4 Report

Concerning the current version of the manuscript I have no further comments to add.